# WARNING: A Wearable Inertial-Based Sensor Integrated with a Support Vector Machine Algorithm for the Identification of Faults during Race Walking

**DOI:** 10.3390/s23115245

**Published:** 2023-05-31

**Authors:** Juri Taborri, Eduardo Palermo, Stefano Rossi

**Affiliations:** 1Department of Economics, Engineering, Society and Business Organization, University of Tuscia, 01110 Viterbo, Italy; stefano.rossi@unitus.it; 2Department of Mechanical and Aerospace Engineering (DIMA), “Sapienza” University of Rome, 00185 Roma, Italy; eduardo.palermo@uniroma1.it

**Keywords:** inertial wearable sensors, artificial intelligence, sports, race walking, support vector machine

## Abstract

Due to subjectivity in refereeing, the results of race walking are often questioned. To overcome this limitation, artificial-intelligence-based technologies have demonstrated their potential. The paper aims at presenting WARNING, an inertial-based wearable sensor integrated with a support vector machine algorithm to automatically identify race-walking faults. Two WARNING sensors were used to gather the 3D linear acceleration related to the shanks of ten expert race-walkers. Participants were asked to perform a race circuit following three race-walking conditions: legal, illegal with loss-of-contact and illegal with knee-bent. Thirteen machine learning algorithms, belonging to the decision tree, support vector machine and k-nearest neighbor categories, were evaluated. An inter-athlete training procedure was applied. Algorithm performance was evaluated in terms of overall accuracy, F1 score and G-index, as well as by computing the prediction speed. The quadratic support vector was confirmed to be the best-performing classifier, achieving an accuracy above 90% with a prediction speed of 29,000 observations/s when considering data from both shanks. A significant reduction of the performance was assessed when considering only one lower limb side. The outcomes allow us to affirm the potential of WARNING to be used as a referee assistant in race-walking competitions and during training sessions.

## 1. Introduction

Exploring the potentialities of artificial-intelligence (AI)-based technologies represents one of the most widespread research fields in the scientific community [1]. Such technologies are increasingly prevalent in society and are also starting to be applied for sports refereeing [2,3,4].

Several sports, such as badminton, race walking or diving, are based on completely subjective referee systems, leading to very frequent controversies on the official results during competitions. In fact, the specific technical gestures and the high dynamics associated with sports are often difficult to perceive by the human eye’s resolution; thus, it is challenging to guarantee fair and equitable games. In addition, the high intra- and interindividual variability in referees has already been demonstrated [5].

Among others, race walking is one of the sports most affected by referees’ judgment. During a race-walking competition, referees must supervise the correct execution of technical gestures, which imply detection of knee-bent (KB) and loss-of-contact (LC) faults [6]. As for the former, the athlete must maintain a full knee extension of the leg in contact with the ground, whereas the latter imposes that at least one foot is always in contact with the ground to avoid running [7]. The correct analysis of the two faults is also made difficult by considering the blindness theory, which occurs when the athletes race in a group [8]. Furthermore, the impossibility to cover all the race circuit with referees represents a further flaw, leading to missed or incorrect disqualifications. For these reasons, the inclusion of race walking among Olympic games is often questioned [9].

Advances in technologies have led to the possibility to objectivize measurements [10]. Specifically, to improve the objectivity in refereeing, innovative technologies based on both wearable sensors and artificial intelligence have been proposed. The study proposed in [11] demonstrated the possibility to achieve an accuracy of approximately 88% in the discrimination of the loss-of-contact fault when using a linear accelerometer placed on the S1 vertebra. Similarly, two sensors with piezoelectric functionalities have been embedded in a shoe sole to measure the sensors’ on/off status, which has been used to supervise the loss-of-contact [12]. The automatic detection of the loss-of-contact was also evaluated in previous studies [13,14], in which the authors showed that using a binary classifier fed with data gathered from an inertial sensor placed on the L5/S1 vertebra can achieve a sensitivity of 82%. A pressure sensor embedded in the athletes’ shoes has been designed and validated for the monitoring of the loss-of-contact by Campoverde-Gárate et al. (2022) [15]. Recently, Caporaso and Grazioso (2020) proposed IART (inertial assistant referee and trainer), an inertial wearable sensor located at the bottom of the vertebral column, for the computation of the loss-of-contact time, achieving an accuracy above 99% [16]. Considering the optimum results, the authors assessed the usability of the IART to support athletes and coaches during training, and as a referee assistant during competition. From this literature overview, it is evident how the proposed technologies only refer to the automatic evaluation of the loss-of-contact fault, neglecting the knee-bent one. To accomplish this, the use of kneepads integrated with strain transducers has been proposed to monitor the extension of the knee during the race walking by [17]. However, such a technology has been considered invasive for the correct execution of the technical gesture, compromising its applicability.

As concerns the possibility to automatically and simultaneously detect both faults, Suzuki and colleagues (2022) verified that a smartphone camera can achieve an accuracy greater than 90% in detecting race-walking faults [18]. However, it is clear how such a methodology cannot be easily applied during official competition. A comparison among several machine learning algorithms fed with data gathered from linear accelerometers and gyroscopes placed on the segments of the lower limb was proposed by the same authors of the present study in Taborri et al. (2019) [3]. The authors reported that the highest accuracy in the classification of both faults was associated with the algorithm based on the support vector machine with a quadratic kernel and trained with data related to the linear acceleration of the shank. Although the system allowed to reach an overall accuracy above 92%, the use of athlete-specific training of the machine learning algorithm could limit its applicability during competition. In fact, the athlete-specific procedure is very time-consuming in terms of both data acquisition and processing [19,20], and it cannot be applied in the daily practices of race walking. However, several studies in different fields, such as gait analysis [21,22], positively evaluated the feasibility of using an intersubject training, which allows realizing a standard dataset of trained parameters to directly use the machine learning algorithm, avoiding the training procedure. 

To the best of authors’ knowledge, no studies in the literature investigated the use of inter-athlete training to implement machine learning algorithms for the automatic detection of both faults during race walking. In this perspective, this paper aims at presenting WARNING, a wearable accelerometer integrated machine learning algorithm trained with parameters gathered from several athletes able to detect both loss-of-contact and knee-bent faults in professional race-walkers. Following this aim, we want to verify if the support vector machine with a quadratic kernel still represents the best-performing algorithm while also considering an inter-athlete training procedure. Finally, we also seek to understand if a single sensor placed on one of the two shanks is sufficient to achieve comparable performance with respect to the use of two sensors, one per each leg. 

## 2. Materials and Methods

### 2.1. WARNING—Hardware and Software Components

WARNING—Wearable Accelerometer for the Recognition of NonInvasive Normed Gesture—is a wearable system for the automatic detection of the technical irregularities of race-walkers. The three main hardware components of the WARNING are:an inertial sensor (Invensense ICM 20948) able to acquire 3D linear acceleration, 3D angular velocity and 3D magnetic field with a sample rate up to 1 kHz;a Bluetooth module 4.0 LE for receiving and transmitting data to a smart device, such as a tablet or smartphone; a processor (Cortex M4).

The roles of the processor are: (i) the acquisition of data from the inertial sensor by using an SPI protocol; (ii) processing the data for the extraction of the main features; (iii) the application of the classification algorithms; (iv) sending the result of the classification for each stride. The hardware is also equipped with a lithium battery, which guarantees an autonomy of 5 h, an on/off switch, a micro-USB port for the sensor charge and an LED for verifying the status of the sensor. A 3D-printed case has been realized to cover the hardware components. Figure 1 shows the hardware components of the sensors and the protective case.

As regards the software components of the WARNING, the sensor firmware was firstly developed. The firmware was written in the C language by using the framework Atollic True Studio. A finite-state machine was used to develop the firmware architecture. For each clock of the processor, the software gathers data from the inertial sensors and preprocesses them with a Butterworth 2nd order low-pass filter with a cut-off frequency set to 20 Hz. Data are then stored each second. From the stored data, the software computes the features used for the classification phase, which are the mean, standard deviation, maximum, minimum, the first two peaks in amplitude of the autocorrelation and the position of the second peak of the autocorrelation, as in [3]. After that, the classification is performed by using a support vector machine with a quadratic kernel trained with standardized parameters that can be set by the operator as the input variable. Raw data are also saved in a .txt file for offline processing. A GUI interface has also been developed for visualizing the results in terms of correct and incorrect strides.

It is worth noticing that, in this study, we used offline processing to test the best-performing classifier and to create the standard parameters for the inter-athlete training procedure.

### 2.2. Participants and Experimental Procedure

Ten expert race-walkers (eight males and two females, 22.2 ± 1.8 years) have been included in the experimental protocol. The dataset was the same used in [3], with the addition of two new racers. Before starting with the experimental protocol, each athlete received a full explanation of the study aims and the tasks to perform, as well as the working principle of the wearable sensors. After this phase, each participant signed an informed consent. All the procedures were conducted at the Istituto di Medicina e Scienza dello Sport of the Italian National Olympic Committee in Rome, Italy. Athletes were included if they had at least five years of expertise in national competitions and if they did not suffer from injuries and/or orthopedic surgery in the last two years. 

Each athlete was sensorized with two WARNING sensors, one per each shank. The position of the sensor was based on the results reported in [3], where the shank was revealed as the most sensitive place for mounting the inertial sensor to achieve the best performance. The sensors were placed by using an elastic strap designed to limit the relative movements between the sensor and the body segment. After the sensorization phase, each athlete was asked to perform a warm-up of 5 min to familiarize themselves with the presence of the sensors. No athlete reported issues related to the wearability of the sensors, which did not interfere with the technical gesture of the race walking.

After the familiarization phase, athletes were asked to race-walk along an ellipsoidal circuit, which was set up to simulate a race circuit. It was composed of a straight path of about 250 m and two curves with a radius of 4 m. A single task consisted of three laps of the circuit performed with the preferred shoes and the preferred race-walking cadence; specifically, the first lap was performed as a regular walking race, the second one by simulating the loss-of-contact fault (LC) and the last one by simulating the knee-bent fault (KB). The experimental protocol consisted of two repetitions of the above-described task. During both repetitions, the data related to the linear acceleration in the three anatomical planes were acquired at 100 Hz and stored for the offline analyses. During the execution of the task, an operator informed the athlete when she/he should modify the type of race walking. An additional sensor, which was synchronized with the two on the athletes’ body segments, was provided to the coach. In particular, the coach oversaw refereeing the type of race walking and he was asked to perform a 180° rotation of the sensor when the athlete performed a transition among the three states of race walking, which were regular, LC and KB. The vertical acceleration gathered from the sensor was then used to find the time instant in which the athlete was changing the race-walking condition. 

### 2.3. Data Analysis

As previously reported, the acquired data has been offline-processed with MATLAB (MathWorks, 2021b, Portola Valley, CA, USA). Linear acceleration signals were low-pass filtered with a second-order Butterworth filter at 20 Hz. The vertical component of the acceleration gathered by the sensor provided to the coach was analyzed to identify the transitions among the three race-walking conditions. Specifically, we evaluated the time instant in which the sign of the vertical acceleration changed in correspondence with the imposed rotation. 

The identification of the time instant in which each athlete changed her/his race-walking condition allowed for the creation of the reference sequence used to associate the data from the WARNING sensors with the relative race-walking condition. Successively, by analyzing the linear acceleration acquired by the WARNING, each walking condition was further partitioned into strides by using a threshold-based algorithm, already validated by [23]. For each stride, a set of seven features was computed; in particular, as for the time domain, we computed the mean, the standard deviation, the maximum and the minimum, whereas for the frequency domain, we calculated the height of the first two peaks of the autocorrelation and the position of the second peak of the autocorrelation. All the features were computed on the linear acceleration curve independently for the three planes. The selection of the previously mentioned features relied on the demonstration that the combination of the time- and frequency-based variables allowed the achievement of a more robust classification, especially when focusing on the periodical time series, such as the movements of race walking [24]. In addition, we included the same features as in [3] to compare the findings between the athlete-specific and interathlete training. Figure 2 includes a scheme of the data analysis.

This procedure led to the construction of an *n x f* matrix, where *n* indicates the number of strides considering all three race-walking conditions and *f* is the number of features. To verify the effects induced by considering both legs (B) or the two sides independently (R and L) for the classification process, we assembled three different dataset subsamples. The first one (B) was composed by a total of 42 features, which are 7 features × 3 planes × 2 body sides, and it was obtained by combining the data gathered from both sensors; the other two dataset subsamples (R and L) were composed by a total of 21 features per each, independently considering the two sensors placed on the right and left leg, respectively. For all the subsamples, we sought to understand their performance in the classification process. 

Data related to the first repetition of the experimental tasks were used for creating the trained model of the machine learning algorithms. The data acquired during the second repetition were instead considered for the validation phase. For the training, we applied an inter-athlete approach considering the data gathered from all the athletes together. Thus, the application of an inter-athlete approach allows avoiding the necessity to create a trained model per each athlete (i.e., athlete-specific training) through the construction a standardized trained model. This approach has also been validated in different movement-recognition analyses, and its results were comparable with the ones obtained through a subject-specific approach [20]. As concerns the validation, a leave-one-out approach was applied, leaving one subject out of the training dataset in turn. 

#### 2.3.1. Selection of the Best-Performing Algorithm

To verify which was the best-performing algorithm when using inter-athlete training, a comparative analysis of 13 machine learning algorithms was conducted. The selected algorithms dealt with the three main categories, which were the decision tree (DT), support vector machine (SVM) and k-nearest neighbors (KNNs). Table 1 shows the tested algorithms.

A brief technical explanation of the selected algorithms is reported in the following.

##### Decision Tree

Decision trees are nonparametric supervised learning methods that aim at creating a decision tree model by applying simple decision rules inferred from the data features. It is considered one of the easiest classifiers to interpret, since it uses a white box model; however, it is also considered unstable, since little variations in data might result in completely different tree constructions [25]. One of the most important parameters to select when implementing a decision tree is represented by the number of maximum allowed splits in the construction of the tree and the split criterion. In our tests, for all the three DT-based algorithms, we selected the Gini diversity index as the split criterion, not allowing surrogate decision splits; the maximum allowed splits were 100, 20 and 4 for the fDT, mDT and cDT, respectively.

##### Support Vector Machines

Support vector machines are among the most widespread machine learning algorithms for the classification of physical activities [26,27]. The classification process starts from the identification of a hyperplane that allows to maximize the possibility to discriminate data points belonging to different classes in an N-dimensional space [28]. The main advantage associated with this type of classifier is the effectiveness in working in high-dimensional space, whereas the main limitation is related to the strong influence of the feature selection, both considering the type and the number [29]. One of the most important parameters to set is represented by the kernel function; in particular, we tested four different kernel functions: linear (lSVM), quadratic (qSVM), cubic (cSVM) and Gaussian (gSVM). In addition, we set the box constraints’ level to 1 and the auto kernel scale mode was disactivated. The multiclass method for the classification was finally chosen as “one-vs-one”, which means the classifier performs the following three comparisons: regular vs. knee-bent, regular vs. loss-of-contact and knee-bent vs. loss-of-contact.

##### K-Nearest Neighbors

K-nearest neighbors are considered one of the simplest classification and regression methods. The procedure for the classification consists in identifying the most likely belonging class by computing and maximizing the distance among different classes [30]. The KNNs typically have a greater predictive accuracy in low-dimensional space in comparison to high-dimensional space; moreover, generally, the results are not easy to interpret [31]. The number of neighbors to be considered when computing the distance and the distance metrics are the two parameters to be selected before the model implementation. Table 2 shows these two aspects for the six tested KNN algorithms. 

For each classifier, we compared the estimated race-walking condition sequence with the one calculated based on the coach refereeing. Successively, a 3 × 3 confusion matrix was computed. From the analysis of the confusion matrix, the overall accuracy of A, the F1 score and the G-index have been calculated, considering the following equations:(1)$$A=\frac{\mathrm{TP}+\mathrm{TN}}{\mathrm{TP}+\mathrm{FP}+\mathrm{TN}+\mathrm{FN}}$$
(2)$$F1-\mathrm{score}=2\cdot \frac{(R\cdot P)}{\left(R+P\right)}$$
(3)$$G=\sqrt{{(1-\mathrm{TP})}^{2}+(1-{\mathrm{TN})}^{2}}$$
where TP, TN, FP and FN are the true positive, true negative, false positive and false negative, respectively; the R and P represent the recall and the precision according to the equations:(4)$$R=\frac{\mathrm{TP}}{\mathrm{TP}+\mathrm{FN}}$$
(5)$$P=\frac{\mathrm{TP}}{\mathrm{TP}+\mathrm{FP}}$$

As for the analysis of the A and F1 score indices, a threshold of 0.80 was assumed for the definition of the optimum classifier [27], whereas the guidelines provided in [20] were taken into account for the G-index as: (i) optimum when G ≤ 0.25; (ii) good when 0.25 < G ≤ 0.70; (iii) random if G = 0.70; (iv) bad if G > 0.70. In our application, we decided to consider a classifier as optimum when all the parameters fell within the optimum range. The above-mentioned metrics were used to compare the three different machine learning algorithms selected in the study.

Finally, per each classifier, the computational load in terms of the prediction speed (PS), expressed as the number of observations per second, was computed through the toolbox Classification Learners provided by Matlab. This parameter allows us to evaluate the feasibility of using the proposed methodology in real-time applications, since the greater the value of the PS, the lower the time necessary for the classification phase. 

#### 2.3.2. Effects of the Number of Sensors

All the previously mentioned analyses were performed independently for the classifiers, fed with data derived from the three different dataset subsamples gathered from the sensors on the two legs (B), and from the sensor on each leg (L and R). To assess the effects induced by considering only one lower limb side on the classification performance, a one-way ANOVA was applied only on the results related to the best-performing classifier. When the ANOVA was found to be significant, a Bonferroni multiple comparison test was applied to investigate the differences among the three dataset subsamples. Before the inference tests, the normality of the data was tested through the Shapiro–Wilk test. For all the tests, the significance level was set to 0.05. Statistical power was computed by using G*power and was found to be equal to 0.89 with a medium effect [32].

## 3. Results and Discussions

A total of 1800 strides, 600 per each race-walking condition, was used for the algorithm training. A total of 500 strides per each condition composed the dataset for the validation phase. 

### 3.1. Best-Performing Classifier

The results related to the performance indices in terms of means and standard deviations are reported in Table 3. In the same table, the results of the prediction speed are shown.

By analyzing the reported results, it is evident that almost the totality of the tested classifiers can be considered as optimum, being associated with an A and F1 score above 80% and a G-index lower than 0.25. This interesting outcome permits the affirmation of the robustness of the implemented algorithms to both type I and type II errors [33]. 

Focusing on the decision tree algorithms, it is evident how these algorithms generally obtained the lowest results; in fact, only the fine decision tree (fDT) achieved index values close to the optimum condition. These results are in line with the ones reported in [3]. These findings can be justified by the well-demonstrated lower effectiveness of the decision tree when considering the inter-subject training procedure [34]. In addition, it is worth noticing that the decision tree is generally considered less effective in physical-activity discrimination in comparison to other machine learning algorithms [35], whereas its use is suggested when seeking to constantly update the model parameters in real-time. 

By focusing on k-nearest-neighbors-based algorithms, we can affirm that the achievement of the optimum performance indices is strongly associated with the selection of the distance metrics and the number of k. In fact, being equal to the selected k, the mKNN showed the lowest results in comparison to the classifiers based on the cosine or cubic distance; moreover, being equal to the distance metrics, the increment of the number of k leads to an increment of the performance (as an example, the A value is 0.90 for the fKNN, 0.88 for the mKNN and 0.77 for the cKNN). Thus, the influence of the selected distance computation method on the performance is evident, suggesting the need to pay attention when selecting this parameter in the KNN implementation, as also discussed in [30]. As concerns the support vector machine, the results are influenced by both the selection of the kernel of the model and the type of the dataset subsampling. More specifically, the linear SVM appeared to generally be the worst classifier, even if falling in the optimum range when considering the subsampling B; moreover, only good performances were found for the other two dataset subsamples. Furthermore, the support vector machine implemented with a quadratic kernel was found as the best-performing and in the optimum range for all the dataset subsamples. By comparing the three typologies of the algorithms, we can confirm, as in [3], that the support-vector-machine-based algorithms reveal themselves as the most appropriate for physical-activities classification. To also consider the variability of the results, Table 4 shows the maximum and minimum values of the three indices for the qSVM, considering all the iterations.

The support vector machine implemented with a quadratic kernel function is the best-performing algorithm when applying an inter-athlete training, as also occurred when using the athlete-specific one, as found in the paper published by Taborri et al. (2019) [3] on the same dataset. In addition, it is worth noticing that the performance of the qSVM trained with an inter-athlete approach is also greater with respect to the ones obtained with the athlete-specific training, as found in Taborri et al. (2019) [3], when considering the F1 score (0.92 vs. 0.89) and G-index (0.09 vs. 0.11), whereas it is comparable when considering the A (0.93 vs. 0.92). These results confirm the ones reported in Goršič et al. (2014) [21], where the authors assessed that the inter-subject procedure could lead to better performance in comparison to the subject-specific one due to the greater robustness of the intersubject variability. Thus, we can affirm the feasibility of applying an inter-athlete training that leads to a significant simplification of the WARNING application. In fact, the time spent to tune the classifier parameter to the specific athlete race-walking patterns can be avoided by using a standard trained model to provide the input to the WARNING software. For this reason, the WARNING software v1.0 has been developed to use the standardized parameters obtained by using data from all 10 athletes. Finally, the standardized parameters can be updated in the case of the enrichment of the training dataset gathering data from the other athletes. 

By considering the PS, the decision-tree-based algorithms are associated with an average value of 40,000 observations/s, whereas the support vector machines and the k-nearest neighbors are associated with 25,000 and 20,000 observations/s, respectively. The results are in line with the expected ones considering the complexity of each algorithm in terms of mathematical computation to perform for the achievement of the classification [36]. Even though the highest value of the PS is associated to the DT algorithms, the value associated with the best-performing algorithm, the qSVM, is suitable enough for real-time application. In particular, the PS for the qSVM was found to be 29,000 observations/seconds. The high values of the PS found for all the classifiers allows for affirming the possibility to use such an approach for real-time applications, since a single observation, which is represented by a single race-walking stride, can be processed in approximately 0.05 ms, guaranteeing an optimal resolution in terms of delay due to the classification phase.

### 3.2. Effect of Sensor Number

For the best-performing classifier, the qSVM, Table 5 reports the true positive (TP), true negative (TN), false positive (FP) and false negative (FN) for each class (regular, loss-of-contact (LC) and knee-bent (KB)). Considering the regular race walking as positive, it is clear how a false-negative rate, which is the number of times that the classifier estimates an irregular stride when the athlete is not faulting their race walking, is also always lower than 5% in the worst case, which is the classifier fed with the L subsampling. A similar rate is also associated with the false positive rate, which is the number of times that the classifier estimates a regular stride when the athlete is faulting their race walking. Such results can be considered sufficient enough to use the WARNING as a supporting tool for referees.

Focusing only on the selected best classifier, and comparing the use of two sensors or only one sensor, statistical analysis revealed the presence of a significant difference due to the dataset subsampling used for the training. In particular, the classifier trained with the data gathered by both sensors (B) is characterized by the highest value of the A and F1 score and the lowest value of the G-index, as in Figure 3. As for the A and F1 score, the *p*-value is equal to 0.05 when comparing B and R, whereas it is lower than 0.01 when comparing B and L. An A *p*-value always lower than 0.001 was found related to the G-index outcomes. By analyzing the histogram, the presence of differences leads to the conclusion that sensorizing both shanks is preferential so to to optimize the prediction accuracy. Conversely, a performance decrease was associated with the use of only one sensor, regardless of the sensorized lower leg side. Even though, according to the selected thresholds (i.e., >80% for the A and F score and <0.25 for the G-index), both L and R models fall in the optimum range, the use of only one sensor is not recommended, especially during an official competition. 

In summary, we can conclude that the WARNING is able to simultaneously and automatically detect irregularities during race walking thanks to the application of a support vector machine algorithm with a quadratic kernel. The findings of this paper permit the consideration of the WARNING as a viable digital tool to support referees during race-walking competitions, as well to monitor the athlete techniques during training. In fact, the system is characterized by: (i) a high level of accuracy, F1 score and G-index; (ii) noninvasiveness; (iii) the simultaneous identification of both faults; (iv) the elimination of athlete-specific training; (v) a suitable level of prediction speed. 

## 4. WARNING Applications and Limits

The WARNING is based on a national Italian patent approved in 2019. The idea behind the realization of a wearable device able to automatically detect faults during race walking is to introduce a supporting tool to help the referees during the decision process of assigning a warning for irregular race-walking techniques. In particular, when the WARNING estimates a fault, i.e., irregular race walking, an alert associated with the specific athlete ID will be sent to the tablet of the referee asking them to check the regularity of the technique before providing the warning. Then, the final decision is upon the referees. From this perspective, it is clear that, even if the findings of the present study reveal the possibility to misclassify 8 out of 100 strides, the WARNING can represent a viable tool for helping, rather than replacing, the referees, especially in the areas of the circuit where it is difficult to check the regularity of the technique. In addition, it should be noted that, due to the biomechanics of the race walking, it is not possible to generate a fault with only one stride. Consequently, the alert to the referee will be sent only after a sequence of more irregular strides, reducing the possibility of an isolated misclassification. Furthermore, the WARNING can also be used during the training sessions in order to improve the race-walking technique of an athlete, who can monitor the regularity of their own movements.

The generalizability of the present study results must be interpreted in light of the following considerations. Firstly, the dataset is composed of race-walkers coming from the same country; thus, the standard parameters found for the inter-athletes can be biased by the race-walking technique, which can be different among different national teams [37]. Secondly, only age-matched race-walkers were enrolled in the study, disallowing us to understand the effects of age on the classification results. Thirdly, the participants did not suffer from any injuries and/or did not undergo any medical treatments before data acquisition; thus, the influence of drugs and/or light injuries on the results cannot be discussed. Finally, it should be also considered that the effects of fatigue can lead to different biomechanics of the race walking and also of the irregular strides. Thus, future studies will concern the enrolment of a greater number of race-walkers, also considering any possible influencing variables (gender, age, nationality, health status), as well as understanding how the results can vary when athletes face fatigue.

## 5. Conclusions

The WARNING is a wearable inertial device able to automatically detect faults during race walking thanks to the use of a machine-learning approach. Experimental outcomes allowed us to affirm the feasibility of its use in an inter-athlete training procedure so to avoid a training phase specific for each athlete, leading to an accuracy greater than 90%. In particular, a quadratic support vector machine algorithm fed with data gathered from sensors placed on both shanks revealed itself as the best-performing in terms of prediction accuracy. The prediction speed value was also found suitable for real-time applications. The WARNING can be used as a supporting tool to help referees in judging race-walking technique during official competition. In addition, the WARNING could be a useful tool during training to monitor the athletes’ technique. 

## 6. Patents

This work is based on the national Italian patent (“Procedimento e dispositivo per rilevare condizioni di marcia”, inventors: Paolo Cappa, Eduardo Palermo, Stefano Rossi, Juri Taborri). 

## Figures and Tables

**Figure 1 sensors-23-05245-f001:**
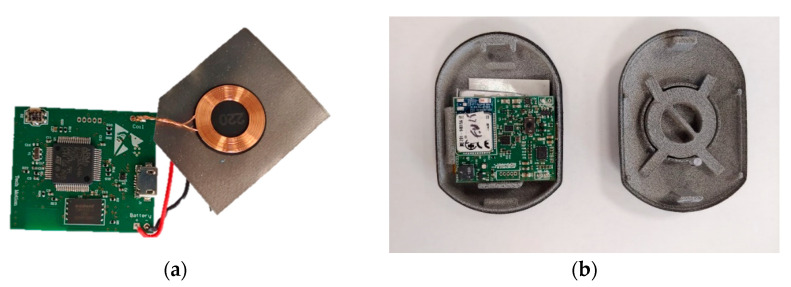
Electrical circuit of the WARNING (**a**) and protective case (**b**).

**Figure 2 sensors-23-05245-f002:**
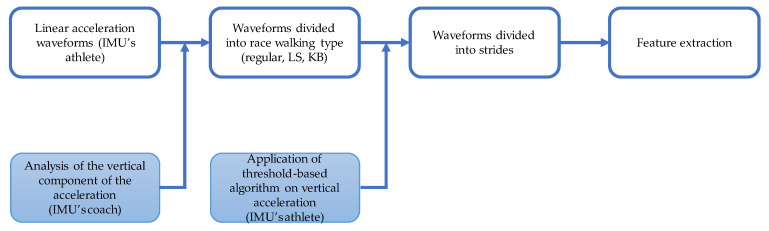
Scheme of the data analysis.

**Figure 3 sensors-23-05245-f003:**
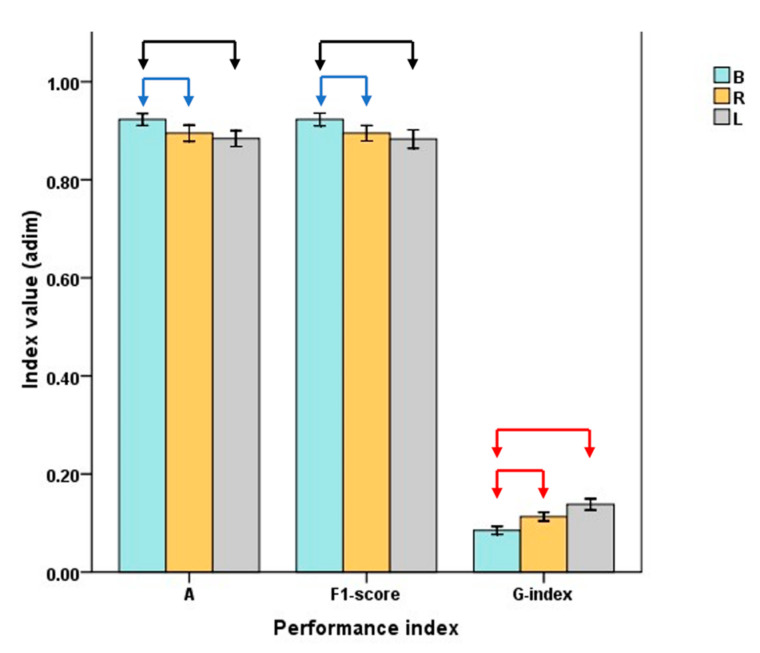
Mean and standard error of the performance index for the quadratic SVM. Arrows indicate the presence of a statistical difference between the type of dataset subsampling. Black, blue and red indicate *p*-values < 0.01, = 0.05 and < 0.001, respectively. B, R and L stand for Both, Right and Left, respectively.

**Table 1 sensors-23-05245-t001:** Tested machine learning algorithms and related parameters for the model construction.

Name	Acronyms	Category
Fine DT	fDT	Decision Tree
Medium DT	mDT	Decision Tree
Coarse DT	cDT	Decision Tree
Linear SVM	lSVM	Support Vector Machine
Quadratic SVM	qSVM	Support Vector Machine
Cubic SVM	cSVM	Support Vector Machine
Fine Gaussian SVM	gSVM	Support Vector Machine
Fine KNN	fKNN	K-Nearest Neighbor
Medium KNN	mKNN	K-Nearest Neighbor
Coarse KNN	cKNN	K-Nearest Neighbor
Cosine KNN	coKNN	K-Nearest Neighbor
Cubic KNN	cuKNN	K-Nearest Neighbor
Weighted KNN	wKNN	K-Nearest Neighbor

**Table 2 sensors-23-05245-t002:** Parameters of the tested KNN classifiers.

Name	Parameter Selection
fKNN	Number of neighbors set to 1. The metric is the Euclidean distance.
mKNN	Number of neighbors set to 10. The metric is the Euclidean distance.
cKNN	Number of neighbors set to 100. The metric is the Euclidean distance.
coKNN	Number of neighbors set to 10. The metric is the cosine distance.
cuKNN	Number of neighbors set to 10. The metric is the cubic distance.
wKNN	Number of neighbors set to 10. The metric is a weighted distance (using the inverse method, where the weight is equal to the inverse of the distance).

**Table 3 sensors-23-05245-t003:** Mean (standard deviation) of overall accuracy (A), F1 score (F1) and G-index (G) for all the tested classifiers and for each type of dataset subsampling. Bold results indicate the best classifier for each dataset subsampling. The last column reports the results for the prediction speed.

	Subsampling B			Subsampling R			Subsampling L			
	A	F1	G	A	F1	G	A	F1	G	PS
fDT	0.84 (0.03)	0.84 (0.03)	0.18 (0.02)	0.84 (0.03)	0.84 (0.03)	0.18 (0.02)	0.80 (0.04)	0.79 (0.03)	0.22 (0.02)	37,000
mDT	0.75 (0.05)	0.75 (0.04)	0.29 (0.05)	0.74 (0.02)	0.74 (0.03)	0.30 (0.04)	0.73 (0.02)	0.73 (0.02)	0.31 (0.04)	39,000
cDT	0.67 (0.09)	0.66 (0.07)	0.38 (0.08)	0.63 (0.07)	0.63 (0.06)	0.46 (0.07)	0.57 (0.10)	0.56 (0.09)	0.59 (0.08)	44,000
lSVM	0.84 (0.02)	0.83 (0.02)	0.19 (0.03)	0.76 (0.02)	0.76 (0.04)	0.27 (0.04)	0.74 (0.03)	0.74 (0.03)	0.29 (0.05)	32,000
qSVM	**0.92** **(0.02)**	**0.92** **(0.02)**	**0.09** **(0.01)**	**0.90** **(0.03)**	**0.90** **(0.03)**	**0.11** **(0.02)**	**0.88** **(0.03)**	**0.88** **(0.03)**	**0.14** **(0.02)**	26,000
cSVM	0.92 (0.04)	0.92 (0.04)	0.09 (0.03)	0.90 (0.04)	0.90 (0.05)	0.11 (0.02)	0.87 (0.04)	0.87 (0.03)	0.17 (0.04)	19,000
gSVM	0.84 (0.03)	0.83 (0.03)	0.22 (0.03)	0.87 (0.04)	0.87 (0.04)	0.17 (0.03)	0.85 (0.03)	0.85 (0.04)	0.19 (0.03)	230,000
fKNN	0.91 (0.05)	0.91 (0.04)	0.11 (0.02)	0.90 (0.05)	0.90 (0.05)	0.12 (0.02)	0.87 (0.04)	0.87 (0.04)	0.15 (0.03)	25,000
mKNN	0.88 (0.03)	0.88 (0.03)	0.13 (0.02)	0.88 (0.04)	0.88 (0.02)	0.13 (0.03)	0.84 (0.04)	0.84 (0.03)	0.18 (0.03)	23,000
cKNN	0.77 (0.05)	0.77 (0.04)	0.26 (0.03)	0.74 (0.04)	0.74 (0.05)	0.30 (0.03)	0.70 (0.09)	0.70 (0.08)	0.33 (0.07)	18,000
coKNN	0.90 (0.03)	0.90 (0.04)	0.12 (0.02)	0.88 (0.04)	0.88 (0.05)	0.13 (0.04)	0.85 (0.05)	0.85 (0.04)	0.17 (0.03)	14,000
cuKNN	0.90 (0.03)	0.90 (0.03)	0.11 (0.01)	0.90 (0.02)	0.89 (0.03)	0.11 (0.02)	0.83 (0.04)	0.83 (0.04)	0.20 (0.02)	19,000
wKNN	0.90 (0.03)	0.90 (0.03)	0.11 (0.02)	0.90 (0.03)	0.90 (0.03)	0.11 (0.02)	0.85 (0.05)	0.85 (0.04)	0.17 (0.03)	21,000

**Table 4 sensors-23-05245-t004:** Maximum (max) and minimum (min) of overall accuracy (A), F1 score (F1) and G-index (G) for all the tested classifiers and for each type of dataset subsampling.

	Subsampling B			Subsampling R			Subsampling L		
	A	F1	G	A	F1	G	A	F1	G
Max	0.95	0.94	0.11	0.92	0.93	0.12	0.90	0.91	0.16
Min	0.88	0.88	0.08	0.87	0.88	0.08	0.86	0.86	0.12

**Table 5 sensors-23-05245-t005:** Mean value of the true positive (TP), true negative (TN), false positive (FP), false negative (FN) for the best-performing classifiers for the three dataset subsamples.

	TP			TN			FP			FN		
	Regular	LC	KB	Regular	LC	KB	Regular	LC	KB	Regular	LC	KB
B	448	454	458	933	951	952	51	33	32	44	38	34
R	436	455	441	914	953	941	70	31	43	56	37	51
L	422	431	442	905	946	920	70	61	50	79	38	64

## Data Availability

Raw data are available under request to the corresponding author.

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
