# Peer review of "WARNING: A Wearable Inertial-Based Sensor Integrated with a Support Vector Machine Algorithm for the Identification of Faults during Race Walking"

_sensors, 2023, doi:10.3390/s23115245_

Round 1

Reviewer 1 Report

There are several English language grammatical issues and spelling issues. All sections should be reviewed carefully for grammatical issues. Example(s) of each:

Spelling:

page 7, line 268: significative -> significant

page 9, line 349: hystogram -> histogram

page 9, line 350: otpimize -> optimize

Grammatical:

page 3, line 125: “we used an offline processing” -> “we used offline processing”

page 8, lines 283-284: “… it is evident as almost the totality of the tested classifiers can be considered as an optimum classifier, …”

Reviewer 2 Report

Title: WARNING: A wearable inertial-based sensor integrated with a 2 support vector machine algorithm for the identification of 3 faults during race walking.

Overall Impression:

(1)  It is a well-formulated paper at addressing how to take bias out of race walking faults.   The overall findings are interesting as they present a way to get reasonable results with the use of 1 sensor using SVM with reasonable accuracy.

(2)  The content was well executed.   The formatting and flow could be improved.  See comments for each section below.

(3) In terms of grammar, it would help to have a full editing on this as there are some places where words that would help with clarification are omitted.  I have noted some of these things below.  However, the entire manuscript has areas like this that would benefit for overall readability.

Introduction

Line 39 – maybe change from ‘well-assessed’ to ‘it is well-established there is a high degree of intra- and inter-individual variability in referees’.

Line 44 – put “the” in front of athlete.

Line 52 – simply cite author and year as opposed to some number – this forces reader to go down to references; put author and year and then cite.

Line 57 – end of sentence ‘loss of contact in’ – it appears this was cutoff.  Loss of contact in what?

Line 58 – as before cite authors as you do in line 59 and 60 or simply note that studies have shown this and put citation at the end of the sentence.  As it stands, it reads awkwardly.

Line 66 – as before cited numeric value, put author and citation at the end.

Line 70 – which paper?  If this is a continuation of the last paragraph, then the starting of a new paragraph seems inconsistent since it goes with previous paragraph. 

Methods

-It seems like 10 is a smaller number and it would be good in results to quantify the inter-athlete variability as well as the within to understand how the classification algorithms are doing.

Additionally, for the test, was it of sufficient length to get results?  For example, in race walking if most errors occur later in the competition when athletes are tired and their motions start to become compromised, how has this set up helped to capture and assess those conditions?

Overall, the description is reasonable.  In some cases, more explanation of your algorithm settings would help.  Additionally, a figure for lines 167-180 to give the reader a sense of what is being described with words might also help.  In description, for example line 192-93, what is the inter-athlete approach?  Many times, in the article you note something but in the nuances of algorithms, this might be helpful to explain fully.  Does this mean all data was aggregated to create the model as opposed to doing it for an individual?  If so, simply state that clearly.

Line 229 – for the one vs one method, can you note again what it is you are predicting?  I am assuming normal, LC and KB, but as you describe your methods, it would be good to link back to the outcomes for the reader, so they know what you are predicting as they are reading.

I see later you split the data into the 3 sensors, but do you also do 3 predictions for each type of walk?  If so note this.

Finally, why not also use the range (i.e., max-min) as a feature.  The min and max by themselves may not be a good feature, but their combination might.  The same would go for frequency domain, the difference in peaks and the heights of them might matter.  You are assuming the raw features are the best predictors of the walking features, when they might not be.

As noted above, there are many instances where grammar for readability might help.  It would be good to have the journal read through for these reasons. 

Results Discussion

For table 3, it would be good to also provide the range (min values and max values for each outcome) and 95% CI.  The means of the two algorithms might be the same, but the ranges would lend insight if one algorithm is subject to creating outlying values (i.e., high min/max).  A 95% CI would also help to display this.

Additionally, what about TP and TN separately?  Some algorithms might do better showing TP or TN, but this is not picked up with the outcome measures chosen.  If for example one algorithm does very well with TP, but not so with TN, then we cannot distinguish that in the reporting of things.  I would report these along with the 3 outcomes you have that use these values.

Line 314-15 – in methods you noted using the inter-athlete training; did you also do the athlete specific?  If so, you should go back and revise the methods to reflect this and also note this in the results.  (see line 200 where you note using this).  If those results are coming from [3] the article (as noted would cite author and year) then note that.  Additionally, if their set up was not the same as yours for the athletes the results might not be comparable.  If that is the case, you should discuss that here briefly.

I think this is noted in 320-28 with universal parameters noted.  If so, then revise the former discussion for clarity.

Figure 2 – can you put the p-values in the figure for clarification? 

Overall: The biggest issue is even with A and F1 scores in the 90’s, this would still represent mistakes in the realm of 10% for TP and/or TN.  So, in this case what is more interesting is also talking about how this could be practically implemented with a human (i.e., observation) or at least in its current form how many people would be potentially dropped from competition if these techniques are used.  I think the authors at least present a compelling case for improvement.  However, it seems like their use practically at this point is not addressed or if race walking went to such a system what would the ramifications be?  Additionally, under ideal conditions we get these results.  What if more race walkers, different ages, those with injuries (i.e., race walkers are going to compete and sometimes be injured or coming off medical treatment), etc.   These things are not addressed in this study and might at least be noted by the authors

See above comments; overall it was fine.  But in some cases words or context was omitted.  In a couple of places it appears as if sentences were cut off.  Go back and review these.

Reviewer 3 Report

Authors of the manuscript titled "WARNING: A wearable inertial-based sensor integrated with a support vector machine algorithm for the identification of faults during race walking" made an inertial-based wearable sensor integrated with a support vector machine algorithm to automatically identify the race walking faults. Authors used two WARNING sensors to gather the 3D linear acceleration related to shanks of ten expert race walkers. The topic is within the scope of the journal. However, the manuscript needs major and sincere revision before accepting:

1) Line 60 and 62, The abbreviation "IART" appeared for the first time within the manuscript. This is a little confusing at first glance. It is suggested to use the full form of it.

2) How did the authors assess the usability of IART to support athletes and coaches during training and as referee assistant during competition?

3) Line 84-86, "To the best of authors’ knowledge, no studies in literature investigated the use of inter-athlete training to implement machine-learning algorithms for the automatic detection of both faults during race walking." This statement isn't completely true. Authors are advised to investigate further on this topic.

4) A few sentences aren't described clearly due to the use of improper prepositions and tenses. It is confusing to readers. It is suggested to be consistent with the tense and prepositions. A grammatical check must be performed on the entire document. This will enhance the readability of this manuscript. (Especially check the Results and Discussions section.)

5) How did the support vector machine-based algorithms reveal themselves as the most appropriate for physical activity classification? And how did the authors compare the three typologies of algorithms?

6) Line 231-239, Will KNN have the same accuracy in high-dimensional space?

7) Line 256-258, "Finally, per each classifier......... has been computed through the toolbox provided by Matlab." The authors should mention which Matlab toolbox was used here.

8) Line 367-375,  Authors are advised to rewrite the conclusion section. Try to focus on the fluency of the sentences. And make sure the argument is reflected in the conclusion section.

Round 2

Reviewer 3 Report

The authors seem to have worked hard on the revision. I have no more comments.